# Exploring Methodologies from Isolation to Excystation for Giardia lamblia: A Systematic Review

**DOI:** 10.3390/microorganisms13081719

**Published:** 2025-07-22

**Authors:** Susie Sequeira, Mariana Sousa, Agostinho Cruz

**Affiliations:** 1ESS, Polytechnic of Porto, rua Dr. António Bernardino de Almeida, 4200-072 Porto, Portugal; sus@ess.ipp.pt (S.S.); ssm@ess.ipp.pt (M.S.); 2REQUIMTE/LAQV, ESS, Polytechnic of Porto, rua Dr. António Bernardino de Almeida, 4200-072 Porto, Portugal

**Keywords:** axenization, encystation, excystation, Giardia lamblia, giardiasis, isolation

## Abstract

Giardia lamblia is a flagellated protozoan and the etiological agent of giardiasis, a leading cause of epidemic and sporadic diarrhoea globally. The clinical and public health relevance of giardiasis underscores the need for robust methodologies to investigate and manage this pathogen. This study reviews the main methodologies described in the literature for studying the life cycle of G. lamblia, focusing on isolation, purification, axenization, excystation, and encystation. A systematic literature review was conducted following the Preferred Reporting Items for Systematic Reviews and Meta-Analyses (PRISMA 2020) statement. Searches were performed in MEDLINE, ScienceDirect, and Web of Science Core Collection databases. A total of 43 studies were included, revealing 58 methods for isolation and purification, 7 for excystation, 2 for axenization, and 5 for encystation. Isolation and purification methods exhibited significant variability, often involving two phases: an initial separation (e.g., filtration and centrifugation) followed by purification using a density gradient for faecal samples or immunomagnetic separation for water samples. Method effectiveness differed depending on the sample source and type, limiting comparability across studies. In contrast, methods used for other life cycle stages were more consistent. These findings underscore the need for standardised methodologies to enhance the reproducibility and reliability of research outcomes in this field.

## 1. Introduction

The etiological agent of giardiasis, *Giardia lamblia* (*G. lamblia*), is a flagellated protozoan parasite that is a major cause of both epidemic and sporadic diarrhoea worldwide [1]. Giardiasis is prevalent and has occurred in outbreaks across the globe since the 1970s. It was listed by the World Health Organization (WHO) as a neglected tropical disease, but it was later removed from the list, reflecting ongoing debate over its global burden and prioritisation [2]. The life cycle of *G. lamblia* comprises two distinct stages adapted to different environmental conditions: an infectious cyst stage and a proliferating trophozoite stage [1,3,4]. The infection is mainly transmitted via the faecal–oral route, resulting from the ingestion of cysts through the consumption of water or food contaminated with faecal matter [1,4]. After ingestion, excystation occurs in the duodenum due to exposure to gastric acid, bile, and pancreatic proteases, resulting in the release of two motile trophozoites per cyst. Trophozoites primarily inhabit the proximal small intestine (duodenum and jejunum) and attach to enterocytes via the ventral disc. During the trophozoite stage, clinical symptoms appear as trophozoites replicate by binary fission into numerous trophozoites. Detached trophozoites pass through the intestinal tract, where encystation is initiated in the small intestine and completed in the colon [1,4]. Cysts are excreted in the faeces and are immediately infectious, remaining viable in the environment until ingestion by a new host, consequently restarting the cycle all over again [3,4]. Axenic in vitro systems enable the study of parasite biochemistry and differentiation mechanisms throughout the parasite life cycle [2]. Excystation and encystation are differentiation processes that occur in response to environmental conditions perceived by the parasite in the host. Both processes can be induced by simulating the conditions found in the human gastrointestinal tract [3]. Excystation and encystation are targets for chemotherapeutic and immunotherapeutic intervention [2]. The clinical and public health burden of giardiasis underscores the need for robust methods to study and manage this pathogen. Diverse techniques have been developed and refined over the years to isolate *G. lamblia* cysts from environmental samples, maintain axenic cultures under laboratory conditions, and understand the mechanisms underlying encystation and excystation. Despite advances in this field, many experimental protocols remain technically complex and poorly standardised. For example, Rice and Schaefer (1981) reported in vitro excystation yields averaging 87% and 70% for cysts from symptomatic and asymptomatic individuals, respectively [5]. In contrast, Boucher and Gillin (1990) observed lower excystation efficiencies, ranging from 10% to 38%, depending on the type of protease used and the origin of the cysts [6]. Furthermore, in vitro encystation protocols are also often inconsistent, as, for example, Fink et al. (2020) achieved only approximately 10% cyst formation using strain GS in an optimised bile–lactic acid medium [7]. Even well-established environmental isolation protocols, such as EPA Method 1623, produce highly variable recovery rates, ranging from less than 1% to over 50%, largely due to differences in sample turbidity and particulate content [8]. The aim of this systematic review was to identify the main methods cited in the literature used throughout the *G. lamblia* life cycle, including isolation, purification, axenization, excystation, and encystation.

## 2. Materials and Methods

The current systematic review was conducted in accordance with the Preferred Reporting Items for Systematic Reviews and Meta-Analyses (PRISMA 2020) statement, which aims to improve the quality and transparency of systematic review reporting [9]. The protocol for this systematic review was registered on the Open Science Framework (OSF) and is available at: https://doi.org/10.17605/OSF.IO/4FP8Y (accessed on 17 July 2025). To formulate the research question and structure the literature search, the SPIDER tool, adapted from Cooke et al. [10], was employed. These frameworks help ensure methodological rigour and comprehensive coverage of the relevant literature. The research question guiding this review was: “What are the predominant methodologies employed for the isolation, purification, excystation and encystation of *G. lamblia*?” (Table 1).

### 2.1. Information Sources and Search Strategy

A literature search was conducted across the MEDLINE, ScienceDirect, and Web of Science Core Collection databases using the following search equation: “(“axenization” OR “isolation” OR “excystation” OR “encystation” OR “purification”) AND (“method”) AND (“giardia”)”. In the ScienceDirect and Web of Science databases, the search was based on title, abstract, or author-specified keywords. In MEDLINE, the available MeSH terms were used. The databases were last accessed on 24 June 2025.

### 2.2. Eligibility Criteria

Studies eligible for inclusion were research articles written in English, Portuguese, or Spanish. These articles were specifically required to assess methods to evaluate the processes of encystation, excystation, isolation, purification, and axenization of *G. lamblia.* Studies lacking explicit specifications of the methods employed were excluded, as were reviews and meta-analyses. No restrictions were imposed regarding the timeframe of the search.

### 2.3. Quality Assessment

A critical appraisal was used for assessing the quality of the studies included in the review, through the development of a checklist adapted from McConn et al. [11]. The checklist included the following quality assessment parameters: clearly specified criteria for sample selection and collection; mention of the source of the samples used; use of controls to validate the experimental procedures; explanation of the methods used and their adaptability to standard laboratory conditions; adequate description of statistics for study replication; assessment of *G. lamblia* viability and functionality; and appropriate discussion of the results obtained. Eight parameters, in the form of questions, were scored on a binary scale as “yes” (criterion met), “no” (criterion not met), or “not applicable/not sure” (Table 2). The total number of “yes” was then summed to assign a quality based on the following scale: 1–2 “yes” = unsatisfactory; 3–4 “yes” = satisfactory; 5–6 “yes” = good; and 7–8 “yes” = excellent.

## 3. Results

The electronic search yielded a total of 234 articles, with 111 entries retrieved from MEDLINE, 63 from ScienceDirect, and 60 from Web of Science (Figure 1). These sets of records were downloaded from each respective database and then integrated into the Rayyan platform [12]. This consolidation served the dual purpose of removing duplicate records and facilitating the retrieval of pertinent articles. Following the elimination of duplicates, a total of 174 studies remained for further evaluation. The titles and abstracts of all identified studies were independently examined by two reviewers, according to predefined inclusion and exclusion criteria. Any disagreements were resolved through consensus discussions. Given the low number of conflicts and their effective resolution, the involvement of a third reviewer was deemed unnecessary. Records that were evidently irrelevant were excluded. In cases where the abstract and/or title did not provide sufficient information to comply with the inclusion criteria, the full text of the report was obtained for thorough evaluation. Consequently, 68 studies were selected for full-text reading, and these were independently assessed by the same two reviewers. Articles that did not meet all inclusion criteria after the full-text assessment (*n* = 25) were excluded from further examination. Figure 1 illustrates and summarises the complete study selection process.

### 3.1. Quality Assessment Results

To minimise the risk of bias, an assessment of the quality of the articles included in the review was conducted through the development of an 8-question checklist. The studies included in the review (*n* = 43) were evaluated based on the question checklist described above. Based on the number of criteria met (“yes” responses), the studies were classified into quality categories (Figure 2). Of the total number of articles included in the review, 79% were classified as good (5–6 criteria met), 16.3% as excellent (7–8 criteria met), and 4.7% as satisfactory (3–4 criteria met). No articles were classified as unsatisfactory (1–2 criteria met), and therefore, no articles were excluded from the review.

### 3.2. Characterisation of Included Studies

The 43 included studies used a wide range of sample types, including water samples (*n* = 23), human faecal samples (*n* = 16), animal faecal samples (*n* = 9), commercial samples (*n* = 11), reference isolates (ATCC 30888, ATCC 30957, and ATCC 50803; *n* = 7), laboratory strains (*n* = 2), and others (rectal swabs, intestinal scrapings, and duodenal fluid; *n* = 4). The water samples in the included studies were collected from various sources (Table 3). Most of the reviewed studies were classified as experimental.

The present systematic review identified a variety of methods used for the isolation, purification, excystation, axenization, and encystation of *G. lamblia*. Many authors often use the terms “isolation” and “purification” interchangeably when referring to methods for obtaining *G. lamblia* cysts or trophozoites. This underscores the very closely related nature of both processes, where isolation typically refers to the initial separation of the organism from its environment, e.g., using a filtration method, and purification involves subsequent steps to further clean the sample, e.g., the sucrose flotation method. To reflect this common practice and ensure comprehensive coverage, both isolation and purification methods were compiled into the same table (Table 4).

To enhance the interpretation of the methodologies retrieved for the isolation and purification of *G. lamblia* cysts or trophozoites, the methods were organised chronologically and grouped into distinct categories according to the principle of the technique employed, when identifiable. The following categories were established:Immunomagnetic Separation (IMS): Uses magnetic beads coated with antibodies for targeted separation.Density Gradient: Involves the use of continuous or discontinuous gradients composed of one or more solutions with varying densities (e.g., sucrose, Percoll) to separate cysts based on buoyancy.Flotation: Use of a single high-density solution (e.g., zinc sulfate) to float the cysts while denser debris settles.Coagulation–flocullation: Employment of chemical coagulants (e.g., aluminium sulfate, calcium chloride) to aggregate suspended particles and cysts into flocs.Sedimentation: Natural or centrifugation-assisted settling of particles based on the difference in sizes or densities.Filtration: A physical separation using mesh, gauze, membranes, or columns, applied mainly to water samples.Fluorescence-based Immunoseparation: Uses fluorescently labelled antibodies to specifically identify and isolate cysts.Mechanical Purification: Procedures involving physical disruption, temperature alternation, or osmotic shock, applied mainly for the selective removal of trophozoites and debris.

### 3.3. Methods for Isolation and Purification of G. lamblia

A total of 58 methods were identified for the isolation and purification of *G. lamblia* and were classified into 15 different groups (Table 4). The initial separation of the organism from the rest of the sample, which often contains large amounts of debris and contaminants, was primarily achieved through a combination of filtration, low-speed centrifugation, and multiple washing steps with solutions such as Hank’s balanced salt solution (HBSS), phosphate-buffered saline (PBS), or distilled water. For further purification of the sample, the two most cited methods were density gradients and immunomagnetic separation (IMS).

Density gradient centrifugation or flotation with sucrose or Percoll^®^ solutions, with densities ranging between 1.10 and 1.275 g/mL, was commonly employed for faecal samples. Polverino et al. compared three techniques for the isolation and purification of *G. lamblia* cysts and concluded that techniques two (sucrose flotation) and three (a combination of centrifugation and sucrose flotation) left the least amount of debris in the purified cyst suspension [45]. Terrones et al. compared four techniques for the isolation and purification of *G. lamblia* cysts and concluded that a two-phase sucrose gradient was most effective [52].

IMS was also quite common in the cited studies, particularly when the sample being analysed was from a water source or when it was intended for further deoxyribonucleic acid (DNA) extraction. IMS purification step involves the selective separation of *Giardia* spp. cysts with magnetic microspheres covered with purified antibodies against the cysts. The bead–organism complex undergoes a dissociation step, usually by either acid or heat dissociation [43,44]. Pinto et al. compared two dissociation procedures (acid and heat) in the IMS method and concluded that acid dissociation was more efficient for *Giardia* spp. cysts [44]. Neto et al. compared calcium carbonate flocculation and membrane filtration, followed or not by an IMS step, and concluded that higher concentrations of cysts were detected when IMS was employed for sample purification [43]. Hsu and Huang compared two purification methods, the Percoll^®^–sucrose density gradient (ICR protozoan method) and IMS (Method 1623) and found that the average recovery efficiency for *Giardia* spp. with Method 1623 was 48.0% greater than that with the ICR protozoan method [36]. In some studies, sucrose gradient flotation was also combined with IMS, as it allows for detailed analysis of the *G. lamblia* genome in clinical stool samples [55].

### 3.4. Methods for the Excystation of G. lamblia

For the excystation of *G. lamblia*, seven methods were identified, as presented in Table 5. In the reviewed studies, excystation was generally performed in two phases: an initial low-pH induction phase using acid solutions (pH ≤ 2.0) and a subsequent excystment phase. The first phase was employed in nearly all cited studies and comprised incubating cysts with aqueous hydrogen chloride (HCl) or a pepsin–acid solution for periods of 5–120 min at 37 °C. The second phase exhibited variation in the cited methods. Some protocols use enzymes such as trypsin dissolved in Tyrode solution or cysteine HCl/ascorbic acid mixtures in HBSS to facilitate excystment, while others neutralise the acid solution with sodium bicarbonate before resuspending the cysts in fresh culture medium [26,29,37,39]. Moreover, the excystment phase often involves specific incubation conditions, such as maintaining the cysts in an inverted position at 37 °C and using various growth media (TYI-S-33 or HSP-3) with a pH ranging from 7.0 to 7.8 [16,29].

### 3.5. Methods for the Axenization of G. lamblia

For the axenization of *G. lamblia*, two methods were identified, as presented in Table 6. Most cited studies involved incubating trophozoites at 37 °C with Keisters’ modified TYI-S-33 medium, using different concentrations and combinations of antimicrobials (such as penicillin, streptomycin, and amphotericin B) to prevent bacterial and fungal contamination [56]. The culture medium commonly has an alkaline pH ranging from 7.0 to 7.2 and is often supplemented with bile salts and bovine serum.

### 3.6. Methods for the Encystation of G. lamblia

For the encystation of *G. lamblia*, five methods were identified, as presented in Table 7. In the reviewed studies, encystation was induced using filter-sterilised TYI-S-33 media supplemented with different concentrations of bovine bile, adjusted to pH 7.8. In the method used by Bielec et al., lactic acid was also included in the encystation medium, as this compound has been shown to stimulate encystation [35]. The inclusion of antimicrobials in the encystation medium was optional.

## 4. Discussion

Among the methods identified for the isolation and purification of *G. lamblia*, density gradient using sucrose or Percoll^®^ and IMS were the most frequently cited. The principle of faecal flotation is based on a gradient produced by the density difference between the solution used and the target sample (parasitic elements). Sucrose or Percoll^®^ solutions are denser than parasitic elements, enabling their isolation at the surface. Factors such as the quantity of examined material, dilution factor used, whether centrifugation is performed, the length of time allowed for flotation, and the type and specific gravity of the flotation solution used can influence the ability to detect parasites and the yield of the flotation process [57].

IMS has demonstrated higher recovery rates, particularly in water samples or when followed by molecular techniques. Nevertheless, its performance is sensitive to sample turbidity, and the method becomes less efficient as turbidity increases, as shown by Ferrari and Veal [32]. While Method 1623, which incorporates IMS, is often cited for its higher cyst recovery efficiency, its high cost and reliance on specific reagents poses a challenge for its application in low-resource settings [36]. Additionally, several factors such as sediment volume, sample volume and turbidity, magnetic material concentration, antibody type, reagents and conditions used, and pH, may affect the efficiency of the IMS method [43,44,55].

Excystation methods identified in this review generally mimic the natural process that occurs within the digestive system of an infected host and usually involves two steps: an acid induction phase at low pH (≤2.0) and a neutralisation step and incubation in nutrient-rich medium. Gastric acid (pH 1.5–3.5) initiates excystation by creating an acidic environment similar to laboratory-induced conditions (pH ≤ 2.0). Upon passage through the stomach, cysts encounter the neutral pH environment of the small intestine (pH 6.0–7.3), similar to the solutions used in vitro to neutralise acid-induced excystation [58,59]. While these in vitro models provide valuable insights, they may oversimplify the complex physiological context of the host and their excystation yield can differ significantly between protocols, ranging from 30% to over 80%, depending on factors such as cyst maturity, duration of acid exposure, temperature, and composition of the neutralisation medium. On the other hand, in vivo excystation, such as the model used by Isaac-Renton et al. [37], allows the assessment of trophozoite recovery under real physiological conditions that may reflect better the true physiological process. However, these models involve ethical and logistical complexities and may also limit the direct extrapolation of findings to human infections, as they differ from human hosts in terms of physiology, immune response, and gastrointestinal environment. Therefore, while in vitro methods allow greater experimental control and reproducibility, in vivo models remain valuable for establishing biological relevance and understanding host–parasite interactions during excystation.

In the axenization step, most studies used Keister’s modified TYI-S-33 medium supplemented with antimicrobials. Antimicrobial agents such as penicillin, streptomycin, and amphotericin B in axenic cultures are essential for the elimination of bacterial and fungal contaminants, which are common in clinical or environmental isolates. However, the optimal combination and concentration of these agents may vary depending on the source of the sample and the sensitivity of the trophozoites to culture conditions. Such variability highlights the importance of adjusting axenization protocols to the biological context of the isolate and the research goal.

Most cited methods for encystation used TYI-S-33 medium with bovine bile and an alkaline pH (around 7.8), which are essential factors for inducing encystation, as together they simulate the conditions found in the small intestine that trigger cyst formation in vivo [57,58,60]. The inclusion of antimicrobials in the encystation medium was optional, which may be due to the specific aims of the studies, such as the need to replicate the natural conditions of the small intestine where natural interactions can occur between *G. lamblia* and the gut microbiota. Barash et al. [17] employed an in vivo encystation protocol, in which mice were orally inoculated with trophozoites to study cyst development under physiological conditions. The study demonstrated that local parasite density plays a key role in triggering encystation in vivo, supporting similar findings from in vitro experiments. These observations highlight the value of in vivo models for capturing the physiological complexity of host–parasite interactions that are difficult to replicate under in vitro conditions.

Regarding *G. lamblia* viability, few studies systematically assessed the integrity and viability of cysts post-purification, excystation, or encystation. Methods such as dye exclusion assays (e.g., trypan blue), fluorogenic vital stains (e.g., resazurin), or molecular viability PCR can be used, but their application was inconsistent across the literature. The presence of non-viable cysts may lead to the misinterpretation of yield and compromise the evaluation of method effectiveness. Hence, performing viability assays is recommended to ensure that isolated cysts remain infectious or metabolically active, particularly if they are to be used for further studies.

Altogether, the findings of this review demonstrate a broad range of methods and highlight significant challenges related to reproducibility, standardisation, and contextual adaptation. Comparative validation studies and the development of clear guidelines are required to support researchers in selecting appropriate, reliable, and resource-appropriate methodologies for working with *G. lamblia* in laboratory settings.

## 5. Conclusions

*G. lamblia* is one of the most common waterborne parasites that infects humans. The cyst stage of this parasite remains viable in the environment until ingestion by a new host and has been extensively identified in various water sources accessed by humans and animals. The present systematic review identified a range of methods used throughout the *G. lamblia* life cycle, including isolation, purification, axenization, excystation, and encystation. The findings reveal significant variation in the design, complexity, and effectiveness of the currently employed methods.

The isolation and purification of *G. lamblia* was the step that showed the greatest variability in the methods used. Sucrose gradient flotation and IMS emerged as the most employed methods; however, their yield is greatly dependent on sample characteristics and operational variables, including turbidity, reagent composition, and centrifugation parameters. Moreover, some of the most used and effective protocols, such as Method 1623, have high cost and technical complexity, which may limit their use in low-resource settings and underscore the need for cost-effective alternatives. In such cases, sucrose and Percoll^®^ flotation, or mechanical purification and filtration methods, offer practical and low-cost alternatives when appropriately optimised.

Excystation and encystation methods are designed to attempt to simulate physiological conditions encountered in the host gastrointestinal tract in vitro. Nevertheless, the reproducibility of these methods remains limited, and their success rates vary significantly across studies. In vitro models are often employed and, while practical, they may not fully replicate the complexity of host–parasite infection. Despite the fact of being rarely used, in vivo studies have provided complementary insights into these biological processes.

Axenization methods require the use of antimicrobials to reduce the microbial contamination of samples. However, there is currently no consensus on the optimal composition or concentration of these combinations, and adjustments are often needed based on the source and growth characteristics of the isolate.

Collectively, these findings show the need for the development and adoption of standardised methodologies that are more accessible, adaptable, and cost-effective. Based on the comparative assessment, IMS and density-based flotation methods remain the most promising for isolation and purification; TYI-S-33-based media are standard for axenization and encystation, while excystation protocols require better harmonisation and validation. Future research should prioritise methodological harmonisation, including the establishment of clearly defined benchmarks and the implementation of reproducibility standards. These efforts will enhance the uniformity of *G. lamblia* research across laboratories and facilitate the establishment of best practices for experimental studies involving this parasite.

## Figures and Tables

**Figure 1 microorganisms-13-01719-f001:**
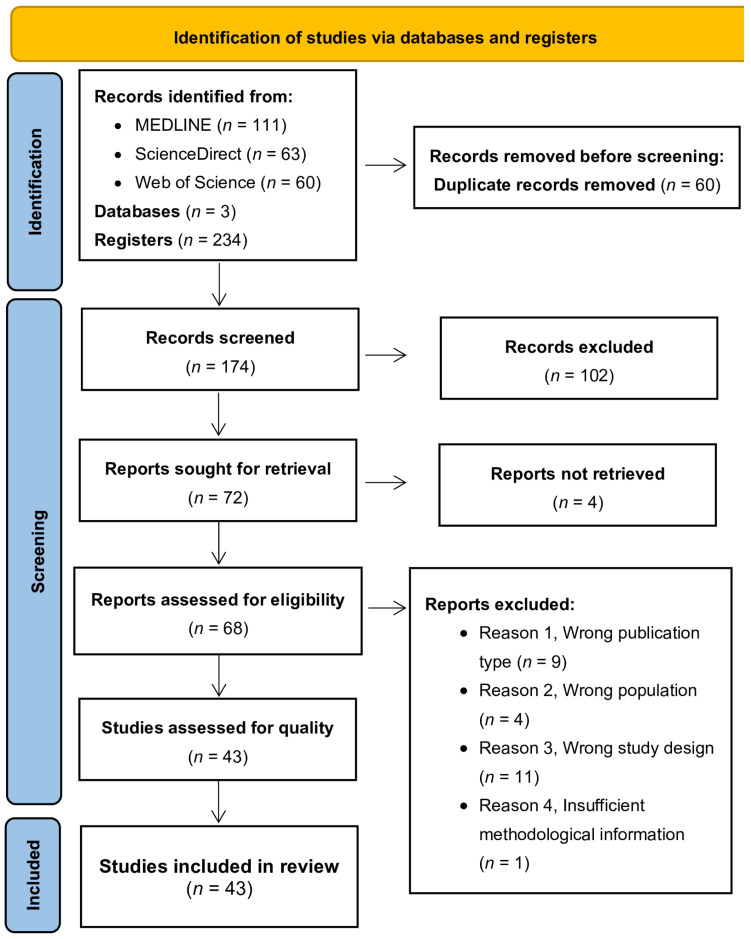
A flowchart of the selection procedure adapted from the PRISMA 2020 statement [9].

**Figure 2 microorganisms-13-01719-f002:**
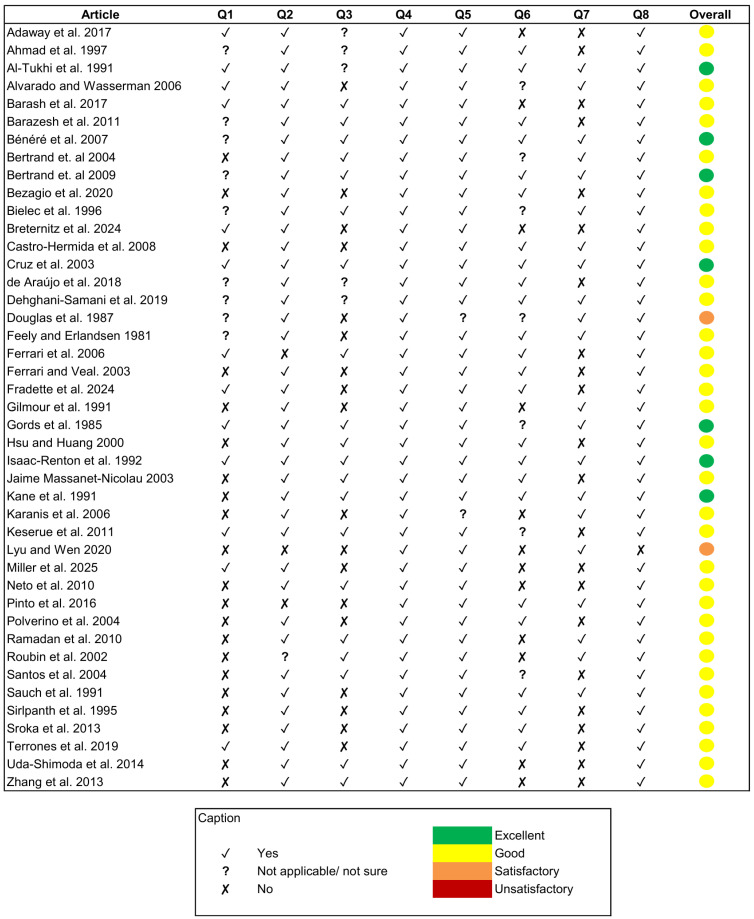
Quality assessment of the reviewed articles [8,13,14,15,16,17,18,19,20,21,22,23,24,25,26,27,28,29,30,31,32,33,34,35,36,37,38,39,40,41,42,43,44,45,46,47,48,49,50,51,52,53,54] considering the questions (Q1 to Q8) formulated in Table 2.

**Table 1 microorganisms-13-01719-t001:** Description of employed SPIDER tool adapted from Cooke et al. [10].

SPIDER Tool	
S—Sample	Studies involving *G. lamblia*
PI—Phenomenon of Interest	Methodologies for isolation, purification, excystation, and encystation
D—Design	Experimental studies
E—Evaluation	Effectiveness, reproducibility
R—Research Type	Methodological studies, experimental research

**Table 2 microorganisms-13-01719-t002:** Questions used as quality assessment criteria for article evaluation.

Questions
1. Did the paper clearly specify the criteria for sample selection and collection, including any inclusion or exclusion criteria?
2. Did the paper mention the source of the samples used (clinical isolates (human or animal), reference isolates, water environment (river, surface water))?
3. Were controls appropriately used to validate experimental procedures?
4. Did the paper explain the methods used (including isolation, purification, axenization, excystation and encystation)?
5. Were the methods described in the paper appropriate and adaptable to typical laboratory conditions?
6. Were the statistics sufficiently described to enable the study to be repeated?
7. Was the viability and functionality of *G. lamblia* assessed?
8. Are the results both justified by the methods provided and discussed in the paper’s discussion section?

**Table 3 microorganisms-13-01719-t003:** Water sample sources in included studies.

Water Source		Number of Studies
Human consumption		8
River		5
Treatment plant	Raw	7
	Sewage sludge	4
	Treated	3
	Sewage effluent	2
	Secondary effluent	1
Wastewater	Abattoir	2
	Treatment plant	4
Dam/Creek/Spring/Well		5
Lake/Pond/Recreational		4
Ultrapure		2
Marine		1

**Table 4 microorganisms-13-01719-t004:** Methods for isolation and purification of *G. lamblia*.

Study Type	Year	Sample	Method	Description	Reference
Experimental	2025	Water (creek)	Immunomagnetic separation (IMS) by US EPA method 1623	The water samples underwent centrifugation (1500 g/10 min). IMS was conducted using the Dynabeads^®^ GC-Combo kit. From the resulting pellet, 0.5 mL was transferred to a tube prefilled with 1 mL of 10 × solution buffer A and 1 mL of 10 × solution buffer B, and distilled water was added to bring the volume to 10 mL. Subsequently, 100 μL of anti-Giardia beads were added, and the sample was rotated for 1 h at room temperature. The tube was placed in a magnetic particle concentrator and gently rocked for 2 min to separate the beads. After removing the supernatant, 1 mL of 1 × solution buffer A was added to each tube. The suspension was transferred into 1.5 mL Eppendorf tubes and subjected to another round of magnetic separation for 1 min. After removing the supernatant and magnet, 50 μL of 0.1 N HCl was added, and the tube was vortexed for 15 s. The tubes were then left to stand vertically for 10 min. The magnet was reinserted to collect the beads; the suspension containing released cysts was transferred to the next tube containing 5 μL of 1 N NaOH.	[42]
	2024	Water (well and drinking water)			[24]
	2024	Water (raw from treatment plant)			[33]
	2018	Water (raw from drinking water treatment plants)			[27]
Experimental	2016	Cysts (EasySeed)			[44]
	2013	Wastewater (treated)			[51]
	2010	Water (river)			[43]
	2008	Water (untreated and treated from treatment plant)			[25]
	2000	Cysts (waterborne) Water (raw and treated from treatment plant)			[36]
	2011	Cysts (waterborne) Water (tap, pond, and river) Wastewater	IMS	A 5 mL sample was passed through the MACS^®^ MS column at a flow rate of 2.05 mL/min using a peristaltic pump. The column was then washed twice with 3 mL of PBS, twice with 3 mL of PBST (PBS with 0.01% Tween 20), and once with 3 mL of PBS at a flow rate of 2.39 mL/min. After removing the column from the magnet, purified cells were recovered by flushing 1 mL of PBS through the column into a 15 mL tube using the provided plunger.	[8]
	2003	Sewage (sludge and anaerobically digested) Bovine manure cysts (University of Arizona)		The sample underwent purification using a commercial kit for the IMS of *G. lamblia* cysts (Dynal^®^). Initially, paramagnetic beads coated with antibodies specific to *G. lamblia* were incubated with the samples. Subsequently, the cyst–bead complexes were isolated from the sample using magnets and subjected to a double washing process. The cysts were dissociated from the beads, and the beads were removed from the purified sample using magnets.	[38]
	2003	Cysts (waterborne) Water (backwash)		A commercial kit (AusFlow IMS kit) specific for *G. lamblia* was obtained from Macquarie Research Ltd. IMS was conducted as described by the manufacturer.	[32]
	2010	Water (tap, secondary effluent, and purified Milli-Q^®^) Cysts (ColorSeed)	IMS with magnetic tube holder	The samples were incubated for 1 h with antibody-coated magnetic beads (Dynabeads^®^ GC-Combo kit). The magnetic-labelled protozoan cells were separated from the liquid matrix using a magnetic particle concentrator. The supernatant was decanted to recover any remaining magnetic particles.	[46]
Experimental	2010	Water (tap, secondary effluent, and purified Milli-Q^®^) Cysts (ColorSeed)	IMS with Magnetic Separation System (MSS)	The samples were incubated for 1 h with antibody-coated magnetic beads (Dynabeads^®^ GC-Combo kit). The magnetic-labelled protozoan cells were separated from the liquid matrix using the MSS. The sample was injected, and the glass flat-sided tube was rinsed four times with filtered PBS. After each rinse, the liquid solution was injected. Once the entire sample was injected, filtered PBS was used to rinse the fluidic system, trapping the magnetic labelled cells and beads in the separation chamber. The supernatant from the first magnetic separation was processed through the MSS again.	[46]
	2002	Cysts (waterborne) Water (river and drinking water)	IMS by AFNOR NF T 90-455	The volume of the concentrated packed pellet was estimated using a calibrated reference and adjusted to 3, 6, or 10 mL based on the IMS kit validation results. Cyst capture occurred over 60–90 min at room temperature using a rotating mixer. The beads were then collected using a magnet in a magnetic particle concentrator during a 2 min shaking procedure. After discarding the supernatant, the isolated beads were resuspended in 1 mL of buffer and transferred to a microcentrifuge tube placed in a magnetic particle concentrator. Following 1 min of rotation at a 90° inclination, the supernatant was removed. Depending on the type of water analysed, the beads were either rinsed with 1 mL of buffer (for raw waters and certain treated waters) or not (for other water types). The antibody–bead complex was dissociated with 50 µL of 0.1 M HCl and vortexed twice for 10 s with a 10 min rest at room temperature. This HCl volume was then transferred to 5 µL of 1 M NaOH and homogenised. Raw waters were further diluted with 55 µL of distilled water.	[47]
			IMS by Chemunex	The procedure was identical to the one cited above, with the sole modification being the doubling of reagents volumes. The antibody–bead complex was dissociated with 100 µL of 0.1 M HCl, vortexed twice for 10 s with a 10 min rest at room temperature, then transferred to 10 µL of 1 M NaOH and homogenised.	
Observational	2019	Faecal (human)	Sucrose gradient	The pellet was mixed with 0.85 M sucrose solution, followed by centrifugation (2300 rpm/10 min/4 °C). The sucrose–water phase was collected, and the pellet underwent 3 washes with distilled water. A gradient tube ranging from 0.85 M to 0.4 M sucrose was prepared. The pellet was added then centrifuged (2300 rpm/10 min/4 °C). The interface was collected, followed by another centrifugation (2300 rpm/5 min/4 °C) to remove sucrose.	[52]
Cross sectional	2017	Faecal (human)	Sucrose gradient	A stool sample was initially diluted with distilled water and filtered through gauze to remove the coarse material. The filtrate was centrifuged (800 g/5 min), and the supernatant was discarded. The resulting pellet was vortexed and divided into two equal portions. Only the second portion underwent the sucrose density gradient method. These stool samples were diluted 1:10 in distilled water, vortexed, and 5 mL of the resulting stool pellet was layered onto 10 mL of 1 M sucrose solution. The mixture was centrifuged (450 g/5 min), and the resulting pellet was subjected to another centrifugation under the same conditions. The final pellet was resuspended in 2.5 mL of distilled water and layered onto 10 mL of 0.5 M sucrose solution, followed by centrifugation (450 g/5 min). The bottom 1 mL of liquid was collected from which 200 μL were dispensed into tubes and stored at 20 °C.	[13]
Experimental	2011	Faecal (human)		The sediments were dissolved in 0.2 M PBS and centrifuged (500 g/5 min), with repeated washing until the supernatant was clear. The sediments were then dissolved in distilled water, poured into tubes containing equal volumes of 1.5 M sucrose solution and centrifuged (1700 g/10 min). The contents of the intermediate phases were then centrifuged again (300 g/5 min/4 °C) to remove sucrose. The obtained sediments were dissolved in distilled water and added to a 0.75 M sucrose solution. The mixture was centrifuged (1700 g/10 min), allowing the cysts to settle while cellulose particles aggregated at the intermediate phase.	[18]
				The sediments were dissolved in 0.5% Tween 80 and centrifuged (500 g/5 min), with repeated washing until the supernatant was clear. The sediments were then dissolved in distilled water and layered onto an equal volume of 1.5 M sucrose, followed by centrifugation (1300 g/10 min/4 °C). The intermediate phase contents were washed 2–3 times by centrifugation (500 g/5 min) then dissolved in distilled water. The solution was added to 0.85 M sucrose and centrifuged (1600 g/10 min/4 °C).	
Experimental	2006	Water (Russia: drinking water and river; Bulgaria: river, lake, well, tap, spring, sewage, and bottled)	Sucrose gradient	Sheather’s sugar solution (500 g sucrose; 6.5 g phenol; 320 mL de-mineralized water) was diluted with 0.1 M PBS (pH 7.2) to solutions A (Sheather/PBS 1:2) and B (Seather/PBS 1:4), supplemented with a few drops of 1% Tween 80. A portion of 15 mL of solution B was layered over 15 mL of solution A. Then, the 10 mL sample suspensions were laid over solution B, and the gradients were centrifuged (1200 g/30 min/4 °C). The resulting supernatants were transferred and washed twice with distilled water. The final pellets were transferred to Eppendorf tubes and stored at 4 °C until use.	[40]
	2003	Faecal (human)		Stools were initially broken up in tap water and filtered through a 300 µm mesh sieve. A 3 mL portion of the resulting faecal suspension was layered onto 3 mL of 0.85 M sucrose and centrifuged (600 g/10 min/4 °C). Cysts located at the sucrose–water interface were aspirated and washed with water. The washed cysts were then layered onto a discontinuous density gradient consisting of two layers: 0.85 and 0.4 M sucrose. After centrifugation (600 g/10 min/4 °C), cysts concentrated at the 0.85–0.4 M sucrose interface were collected and subjected to another wash. Purified cysts were resuspended in distilled water and stored at 4 °C.	[26]
	1991	Faecal (human)		Stool samples were shaken with CDW for 30 min and allowed to settle. The supernatant was collected and layered onto a gradient of 0.4 M and 0.85 M sucrose solutions, followed by centrifugation (600 g/10 min). The material at the water/sucrose interface was aspirated and washed twice with CDW. Subsequently, the cyst pellet was resuspended in a mixture of 5 mL CDW and 7 mL ethyl ether, vortexed, and centrifuged again (600 g/10 min).	[15]
Comparative	1991	Faecal (human)		Faecal samples were diluted either 1:10 or 1:20 with distilled water and passed through a series of sieves (pore size 50 µm) under negative pressure. The filtered faecal suspension was layered onto a cold discontinuous sucrose gradient consisting of 10 mL of 1.02 g/mL, 25 mL of 1.06 g/mL, and 10 mL of 1.18 g/mL sucrose solutions. After centrifugation (400 g/10 min), the middle portion at 1.06 g/mL density was collected and diluted 4 times with distilled water. Cysts were then centrifuged (400 g/10 min) and washed 3 times with distilled water.	[34]
Observational	2019	Faecal (human)	Sucrose flotation	A saline solution was added to the pellet and centrifuged (1500 rpm/5 min). Subsequently, 4 mL of PBS and 4 mL of sucrose solution (density 1.275 g/mL) were added. The mixture was then centrifuged (2300 rpm/10 min). The sucrose sediment phase was aspirated, followed by 3 washes with distilled water to remove sucrose.	[52]
Experimental	2019	Faecal (human)		A stool sample was diluted in 100 mL of tap water and filtered through a 300 µm filter. A 3 mL aliquot of the filtered faecal suspension was mixed with 3 mL of 0.85 M sucrose and centrifuged (2000–3000 rpm/10 min). The cysts at the sucrose–water interface were aspirated and washed 3 times with water. The purified cysts were layered onto a discontinuous density gradient consisting of 3 mL layers of 0.85 M sucrose. After centrifugation (2000–3000 rpm/10 min), cysts concentrated at the sucrose interface were collected and washed again.	[28]
	2017	Faecal (animal)		Samples were suspended in 10 mL tap water, homogenised, and filtered through a tea strainer. Then 5 mL of faecal solution was layered onto an equal volume of chilled 0.75 M sucrose and centrifuged (400 g/5 min). A total of 2 mL of cyst-containing solution was removed from the water–sucrose interface with a sterile transfer pipette, quantified with a hemacytometer, and diluted to 1000 cysts/mL with tap water.	[17]
	2014	Faecal (human)		A faecal suspension sample was layered on 2.5 mL of 1 M sucrose (density 1.11 g/mL) and centrifuged (400 g/15 min/20 °C). Cysts concentrated at the water–sucrose interface were collected, washed by resuspending in 4 mL of normal saline, and sedimented by centrifugation (600 g/10 min). After removing the supernatant, the cysts were resuspended in normal saline and counted using a hemacytometer.	[53]
	2011	Faecal (human)		The sediments were dissolved in a 0.5% Tween 80 solution and centrifuged (700 g/5 min). This process was repeated until the supernatant was clear. The sediments were then collected, prepared into suspensions with appropriate concentrations by adding distilled water and were decanted into tubes containing equal volumes of 0.85 M sucrose solutions. The tubes were centrifuged (500 g/20 min). The thin layer formed between the distilled water and sucrose phases was centrifuged 2–3 times (500 g/5 min) to remove the sucrose. The resulting sediments were then dissolved in distilled water to create suspensions.	[18]
Experimental	2003	Sewage (sludge and anaerobically digested) Bovine manure cysts (University of Arizona)	Sucrose flotation	A volume of 10 mL of 0.01% Tween 20 was added to the spiked biosolid sample and vortexed for 60 sec. A solution of sucrose (density 1.18 g/mL) was injected underneath the sample, followed by centrifugation (1050 g/10 min). The top 10 mL of the sample, the interface between the layers, and the top 10 mL of sucrose were collected and centrifuged (1050 g/10 min).	[38]
	1997	Water (raw and treated from treatment plant)		Samples were initially filtered through a polypropylene cartridge filter (pore size 1 µm). The filters were cut and teased apart, and each half of the filter matrix was placed into double-layered plastic bags. These were subjected to stomaching for 10 min in 1 L of 0.1% Tween 80 in distilled water. After allowing the mixture to settle for 24 h at 4 °C, the supernatant was centrifuged (1500 g/10 min). Subsequently, the samples underwent sucrose flotation using a cold sucrose solution (density 1.18 g/mL), followed by centrifugation (1000 g/5 min) and washing with PBS.	[14]
	1992	Faecal (human and animal)		A faecal suspension sample was layered on 2.5 mL of 1 M sucrose (density 1.11 g/mL) and centrifuged (400 g/15 min/20 °C). Cysts concentrated at the water–sucrose interface were collected, washed by resuspending in 4 mL of normal saline, and sedimented by centrifugation (600 g/10 min). After removing the supernatant, the cysts were resuspended in normal saline and counted using a hemacytometer.	[37]
	1991	Faecal (mongolian gerbils)		Flotation of the faecal slurry over 1.0 M sucrose.	[49]
	2006	Faecal (human)	Combination of filtration and sucrose flotation	The samples were diluted with distilled water and filtered through gauze to remove coarse material. The filtrate was then centrifuged (800 g/5 min) and the supernatant was discarded. The resulting pellet was washed, resuspended in distilled water, and divided into four aliquots of 5 mL. Each aliquot was placed over 3 mL of cold 0.85 M sucrose and centrifuged (600 g/10 min). The interfaces were recovered, mixed, and diluted 25-fold with distilled water. The diluted mixture was vacuum filtered through a 5 μm cellulose acetate membrane. The filter was washed, and the cysts collected on the membrane were sedimented by centrifugation (800 g/5 min). The purified cysts were stored at 4 °C with antibiotics (1.000 U penicillin and1 mg/mL streptomycin).	[16]
Experimental	2006	Water (Russia: drinking water and river; Bulgaria: river, lake, well, tap, spring, sewage, and bottled)	Combination of filtration and sucrose flotation	The water samples underwent filtration using a polypropylene cartridge filter with 1 mm porosity. The filters were cut open and washed twice in 0.1% Tween 80 solution, followed by centrifugation (2100 g/10 min/4 °C). These processes were repeated twice. The resulting pellets were layered over an equal volume of sucrose 2.5 M and centrifuged (300 g/13 min/4 °C). The supernatants were carefully transferred and washed twice with distilled water (2100 g/10 min/4 °C). Further purification of the samples, using 1.5 M sucrose, was performed when necessary.	[40]
Comparative	1991	Water (tap, raw, recreational, and marine) Sewage effluents		Shredded filters were washed with 0.1% Tween 80 using a customised washing machine. The eluate was concentrated to 20 mL, either by centrifugation or sedimentation, and the pellet volume was recorded. Cysts were further concentrated by sucrose density flotation, and the fluid above the pellet, along with the interface, was aspirated. It was then diluted in cold distilled water (CDW) and concentrated to a minimal volume (~ 1 mL).	[34]
Experimental	2004	Faecal (human and dog)	Centrifugation	1. Suspension of faecal matter in Telemann solution (50 mL of formalin 40% and 5 g of NaCl in 950 mL of distilled water) with 2 mL of ether, followed by centrifugation (1500 rpm/5 min). The supernatant was discarded, and the pellet was washed with PBS (pH 7.2).	[45]
			Sucrose flotation	2. Suspension of faecal matter in PBS, followed by centrifugation (1500 rpm/5 min). PBS was added and resuspended by vortex. Then, 4 mL of sucrose solution (density 1.275 g/mL) was added and centrifuged (1500 rpm/5 min). The sucrose phase and the pellet were transferred, and 2 volumes of PBS were added. The cysts were concentrated by centrifugation (2000 rpm/5 min). The sediment was resuspended with PBS in a final volume of 0.5 mL and stored at 4 °C.	
			Combination of centrifugation and sucrose flotation	3. Application of 1 plus 2 on the resulting pellet.	
Experimental	2004	Water (sludge)	Combination of ether clarification procedure (ECP) and sucrose flotation	For ECP, aliquots of filtrate were diluted (1:3) with 1% Tween 80 solution. The pellets were retrieved by double centrifugation (1500 g/15 min), followed by further concentration using ECP (1500 g/10 min) after manual agitation (30 s). For sucrose flotation, following double centrifugation, a saturated sucrose solution (1.20 g/mL) was added to the pellets and centrifuged again (1500 g/15 min). The superficial layer (3 mL) was collected and transferred, and this procedure was repeated.	[48]
	2011	Faecal (human)	Percoll–sucrose gradient	The sediments were dissolved in distilled water and centrifuged (500 g/5 min), with repeated washing until the supernatant was clear. The sediments were then dissolved in distilled water and added to equal volumes of 1 M sucrose solution, followed by centrifugation (500 g/10 min). The middle phase contents were collected and washed 2–3 times with distilled water. The suspension was then added to two Percoll solutions (densities of 1.05 and 1.09 mg/mL) and centrifuged (500 g/20 min). The contents of the phase created between the two Percoll layers were collected, washed 2–3 times, and mixed with distilled water.	[18]
	2009	Cysts (waterborne) Faecal (human and dog) Wastewater (abattoir and treatment plant)		A 4 mL pellet of sewage sample was resuspended in 6 mL formalin 10% and 3 mL ethyl acetate, followed by vortexing. Centrifugation (500 g/5 min) resulted in the formation of 4 layers, with the top 3 being decanted. The resulting pellet was washed with deionized water, centrifuged again and used for Percoll–sucrose flotation. It was resuspended in 20 mL of deionized water and layered onto 30 mL of Percoll–sucrose solution (density 1.10 g/mL). After centrifugation (1050 g/10 min), the aqueous suspension and 5 mL of Percoll–sucrose solution under the interface were collected. The collected material was washed with an equal volume of deionized water by centrifugation (2000 g/10 min). Two additional washings were carried out, and the pellet was conserved in deionized water at −80 °C.	[21]
	2004	Cysts (waterborne) Wastewater (raw)			[20]
Experimental	2000	Cysts (waterborne) Water (raw and treated from treatment plant)	Percoll–sucrose gradient	After resuspending the pellet to a packed volume of up to 0.5 mL, it was vortexed with eluting solution to reach a final volume of 20 mL. The vortexed suspension was then layered with 30 mL of Percoll–sucrose flotation solution (density 1.11 g/mL) and centrifuged (1050 g/10 min). Following centrifugation, the top 20 mL particulate suspension layer, the interface, and 5 mL of the Percoll–sucrose below the interface were transferred. Eluting solution was added to reach a final volume of 50 mL, and the mixture was centrifuged again (1050 g/10 min). The resulting pellet was resuspended by vortexing.	[36]
	1996	Reference isolate (ATCC 30957) Water (recreational and tap)			[23]
	1991	Faecal (mongolian gerbils)		Crude isolates obtained via sucrose flotation were layered onto a Percoll density gradient ranging from 1.01 to 1.03 g/mL. The sample band was allowed to sediment at room temperature for 1.5 h. After microscopic examination, uncontaminated fractions were collected and washed 3 times with distilled water by centrifugation (650 g/2 min) to remove Percoll. The cysts were stored at 4 °C in distilled water.	[49]
	2009	Cysts (waterborne) Faecal (human and dog) Wastewater (abattoir and treatment plant)	Percoll–sucrose flotation	A 4 mL pellet of sewage sample was resuspended in 6 mL of formalin 10% and 3 mL of ethyl acetate, followed by vortexing. Centrifugation (500 g/5 min) resulted in the formation of 4 layers, with the top 3 being decanted. The resulting pellet was washed with deionized water, centrifuged again, and used for Percoll–sucrose flotation. It was resuspended in 20 mL of deionized water and layered onto 30 mL of Percoll–sucrose solution (density 1.10 g/mL). After centrifugation (1050 g/10 min), the aqueous suspension and 5 mL of Percoll–sucrose solution under the interface were collected. The collected material was washed with an equal volume of deionized water by centrifugation (2000 g/10 min). Two additional washings were carried out, and the pellet was conserved in deionized water at −80 °C.	[21]
	2004	Cysts (waterborne) Wastewater (raw)			[20]
Experimental	2020	Faecal (animal)	Zinc sulfate flotation	Faeces samples were suspended in water and filtered through 60-mesh and 200-mesh sieves. The resulting filtrate was centrifuged (1500 g/10 min) and the supernatant was discarded. To the sediment, a 33% zinc sulfate solution of equal volume was added for resuspension, followed by centrifugation (1200 g/15 min). The supernatant was transferred and diluted with 4 volumes of water before another round of centrifugation (500 g/10 min). The last four steps were repeated twice to purify the cysts, and the precipitated cysts were resuspended with PBS.	[41]
Observational	2019	Faecal (human)		A solution of zinc sulfate (density 1.18 g/mL) was used, causing the residue to settle at the bottom of the tube. The supernatant was then centrifuged (2300 rpm), followed by 3 washes with distilled water to remove zinc sulfate.	[32]
Experimental	2017	Faecal (human)		After filtering a homogeneous mixture of faeces and water, the strained faecal suspension underwent centrifugation, and the sediment was then resuspended in a zinc sulfate solution (density 1.18 g/mL). The suspension underwent centrifugation again, which resulted in the flotation of the cysts.	[53]
	2013	Cysts (waterborne) Water (tap, reservoir, river, and sewage) Wastewater (treatment plant)	Coagulation–flocculation	Each sample underwent rapid mixing at 180 rpm for 2 min, followed by slow mixing at 40 rpm for 15 min, and then settling for 30 min. The coagulant (polyaluminium chloride) was added during the rapid mixing period. During coagulation, metal ions hydrolysed into polymeric substances and adsorbed colloidal particles, forming visible flocs which entrapped the cysts. The supernatant was discarded, leaving approximately 15 mL of flocs/L. The pH of the metal hydroxide solution was then adjusted to 2.0, triggering an acid–base neutralisation reaction that dissolved the flocs and released the cysts. The samples containing the cysts were filtered through membranes (pore size: 0.45 μm). The cysts were trapped on the membrane and rinsed with PBS at pH 3.0 and pH 8.0.	[54]
	2010	Water (river)		A solution of CaCl_2_ and a solution of NaHCO_3_ were added separately to the sample (pH 10.0). Samples were kept overnight at room temperature. The supernatant was aspirated, and the precipitate was dissolved with 10% sulfamic acid. The suspension was centrifuged, and the supernatant was aspirated. The pellet material was centrifuged again.	[43]
Experimental	2006	Water (Russia: drinking water and river; Bulgaria: river, lake, well, tap, spring, sewage, and bottled)	Coagulation–flocculation	An aqueous aluminium sulfate solution (16 mg Al^3+^/L) was added to each water sample, with pH adjusted to 5.4–5.8. The samples were left overnight in the dark at room temperature to allow flocculation and precipitation. The resulting pellet was centrifuged (2100 g/10 min/4 °C) and then resuspended with a lysis buffer (8.4 g citric acid monohydrate; 17.64 g trisodium citrate dihydrate; distilled water up to 100 mL; pH 4.7). After settling with the lysis buffer for 1 h at room temperature, the pellet underwent a double washing with distilled water.	[40]
	2006	Cysts (waterborne) Wastewater (abattoir)		A totally of 10 mL of 1 mol CaCl_2_ and 10 mL of 1 mol NaHCO_3_ was added to the water samples (pH 10.0). The samples were centrifuged and washed with a buffer solution and 0.05% Tween 80 solution. The samples were prefiltered through a series of 50 and 38 µm stainless steel mesh filters.	[31]
	2010	Water (river)	Filtration	The samples were filtered through mixed cellulose ester membranes (with 3 mm porosity). The membranes were scraped and manually rinsed with eluting solution (0.1% Tween 80). The resulting liquid was centrifuged and the pellet was rinsed with Milli-Q^®^ water.	[43]
	2004	Water (sludge)		Filtration with a 1 mm^2^ plastic sieve to remove large debris.	[48]
	2003	Cysts (waterborne) Water (backwash)		Water concentrates were centrifuged (13,000 g/10 min) and the supernatant was discarded. Samples were resuspended to the initial volume in mAb buffer (2 mM of tetrasodium pyrophosphate, 2% (*w*/*v*) bovine serum albumin, and 0.05% (*v*/*v*) Tween 80, with pH = 8). Samples were filtered (38 µm stainless steel mesh filter) in a Swinnex filter unit with a syringe.	[32]
	2002	Cysts (waterborne) Water (river and drinking water)		The water samples were filtered using Envirocheck cartridges (pore size of 1 µm). Filtration involved mechanical shaking at 600 rpm for 10 min, which was repeated twice. During the second agitation phase, the cartridge was positioned at a 45° angle.	[47]
	1987	Water (raw from treatment plant)		A stool sample was diluted 1:20 in CDW or phosphate-buffered saline (PBS), filtered through cheesecloth, and chilled in ice-water (4 °C). Sephadex G-50 columns were prepared by suspending 30 g of Sephadex G-50 in 1 L of PBS (pH 7.2) for 3–5 days at room temperature. The diluted stool sample was layered onto the packed Sephadex column, and the cysts were washed with 5 volumes of CDW or PBS. Fractions of 50 mL were collected, and cysts were identified microscopically. Fractions containing cysts were washed 4–5 times in CDW or PBS by centrifugation (1000 rpm/5 min/4 °C).	[29]
Observational	2019	Faecal (human)	Sedimentation	Water or other low-density liquids, such as physiological saline solution, were used to recover the evolutionary microscopic forms of the parasites. These forms settle at the bottom of the tube, deposited there due to their density.	[52]
Experimental	2017	Faecal (human)		A homogeneous mixture of faeces and water was filtered and then left to sediment for 12 h. The resulting sediment was then resuspended in distilled water and allowed to settle again. The washing step was repeated multiple times until the sample was considered cleaned.	[53]
	2003	Sewage (sludge and anaerobically digested) Bovine manure cysts (University of Arizona)		A volume of 25 mL of 0.1 M PBS was added to the spiked biosolid sample and vortexed for 60 s. An additional 25 mL of PBS was added, and the sample was left to stand for 60 min at room temperature. The supernatant was collected and centrifuged (1050 g/5 min).	[38]
	1995	Faecal (human) Rectal swabs (dogs)		Each specimen was emulsified in 0.85% saline solution and filtered through gauze to eliminate large particles. Ten volumes of distilled water were added to the filtrate, which was then centrifuged (400 g/10 min/room temperature). The sediment containing the cysts was rewashed twice.	[22]
	2006	Cysts (waterborne) Wastewater (abattoir)	Fluorescence-based immunoseparation	The cyst wall-specific IgG1 mAb G203, conjugated to fluorescein isothiocyanate, were used for sample prestaining at concentrations of 2 mg/mL and 4 mg/mL, respectively, in buffer. The samples were vortexed and incubated in the dark at 4 °C for 15 min. Fluorescence-activated cell sorting (FACS) was performed on a BD FACSCalibur-Sort equipped with a 488 nm argon ion laser for excitation. The FACSCalibur was equipped with a SortStage attachment, which allowed for the capture of sorted cells onto 13 mm (pore size: 0.8 µm) isopore polycarbonate membranes.	[31]
Experimental	2020	Faecal (human)	Modified Ritchie Method	One gram of faeces was diluted in 0.85% saline solution, then filtered through gauze and centrifuged (1200 g/5 min). Several types of water (Milli-Q^®^ water, distilled water, deionized water, and injection water) were added along with ethyl ether (6:4 mL). After vigorous stirring, the samples were centrifuged (1200 g/2 min). The final sediment received the amount of each respective reagent necessary to complete 150 µL. A portion of 50 µL was used for quantification in a haemocytometer.	[22]
	2020	Intestines of host (animal)	Mechanical purification	The intestines of the host animals were dissected into segments and transferred into TYI-S-33 medium, where they were chilled on ice for over 30 min. The suspension was centrifuged (1000 g/1 s), and the supernatant was transferred to a new tube, which was then centrifuged (750 g/5 min). After discarding the supernatant, the sediment was resuspended with TYI-S-33 and kept at 37 °C for 30 min. The TYI-S-33 was replaced with fresh TYI-S-33 and incubated at 37 °C for another 30 min, repeating this step once more. Subsequently, the tube was chilled on ice for 30 min, centrifuged (2000 g/5 min), and the supernatant was discarded. The trophozoites were collected from the pellet and resuspended in PBS.	[41]
	1991	Cysts produced in vitro		To isolate cysts, trophozoites were removed by hypotonic lysis in distilled water for 1–24 h at 4 °C, followed by successive washing in distilled water to remove debris.	[39]
	1981	Intestinal scrapings (rats)		The scrapings were suspended in 10 mL of HBSS (pH 7.2) and aspirated several times to break up tissue fragments and dislodge trophozoites. The suspension was centrifuged (200 g/1 min), and the supernatant was further centrifuged (200 g/10 min) to pellet the trophozoites. The bottom 4 mL of each tube, including the trophozoite-enriched pellet, was transferred to a Petri dish and incubated at 37 °C for 20 min to allow trophozoites to adhere to the dish surface. The suspension was removed, leaving the attached trophozoites undisturbed. The dishes were rinsed twice with warm HBSS (37 °C), tilting the dish to allow 1–2 mL of HBSS to wash over the surface before removal. The surface was covered with 2 mL of fresh HBSS (37 °C).	[30]
Experimental	1981	Intestinal scrapings (rats)	Mechanical purification	Petri dishes with attached trophozoites were initially incubated at 37 °C (cycle 1; 37 °C) to promote attachment. Subsequently, the dishes were incubated at 4 °C for 20 min (cycle 1; 4 °C) and 2 mL of cold HBSS was used to rinse and collect detached trophozoites, leaving intestinal debris adherent to the dish. The collected washings were transferred to clean Petri dishes, and the procedure was repeated for two additional cycles alternating between 37 °C and 4 °C to purify the trophozoites. After each cycle, reattached trophozoites were rinsed with 1 to 2 mL of HBSS.	[30]

Caption: CaCl_2_—Calcium Chloride; CDW—Cold Distilled Water; ECP—Ether Clarification Procedure; FACS —Fluorescence-Activated Cell Sorting; HBSS—Hank’s Balanced Salt Solution; HCl—Hydrogen Chloride; IMS—Immunomagnetic Separation; MSS—Magnetic Separation System; NaCl—Sodium Chloride; NaHCO_3_—Sodium Bicarbonate; NaOH—Sodium Hydroxide; PBS—Phosphate-Buffered Saline; PBST—PBS with 0.01% Tween 20; SFM—Sucrose Flotation Method; UK—United Kingdom; USA—United States of America.

**Table 5 microorganisms-13-01719-t005:** Methods for excystation of *G. lamblia*.

Study Type	Year	Sample	Method	Excystation Medium	Description	Reference
Experimental	2009	Cysts (waterborne) Faecal (human and dog) Wastewater (abattoir and treatment plant)	Modified method of Bingham and Meyer	Pepsin–acid solution with 25 mM NaHCO_3_; 12 mM KCl; 40 mM NaCl; 6 mM CaCl_2_; and 1500 U/mL pepsin (pH 2.0)	A cyst suspension was added to 1 mL of pepsin–acid solution and incubated at 37 °C for 1 h. The suspension was then centrifuged (800 g/3 min), and the resulting pellet was washed twice with PBS and resuspended in TYI-S-33 at 37 °C.	[20]
	2006	Faecal (human)		Aqueous HCl (1:10, pH 2.0)	A cyst suspension was mixed with excystation medium and incubated at 37 °C for 1 h. After centrifugation (600 g/5 min), the pellet was washed with water, recentrifuged, and then resuspended in 0.5 mL of HSP-3 at 37 °C.	[16]
	1991	Faecal (human)		Aqueous HCl (1:10, pH 2.0)	A cyst suspension was mixed with excystation medium and incubated upright at 37 °C for 1 h. After centrifugation (600 g/10 min), the pellet was collected, inoculated into prewarmed modified TYI-S-33 medium, and incubated overnight in an inclined position to allow complete hatching.	[15]
	2003	Faecal (human)	Method of Schupp et al.	Dilution of 100 mM cysteine–ascorbic acid solution in Eagle’s minimal essential medium solution in HBSS	Cysts were inoculated with warm excystation medium (37 °C) and incubated for 15 min at 37 °C. The cysts were centrifuged, and the resulting pellet was resuspended in TYI-S-33. The tubes were incubated at 37 °C and examined microscopically for 48 h.	[26]
	1995	Faecal (human) Rectal swabs (dogs)	Method of Sirlpanth et al. [50]	0.4 pepsin in HCI (pH 1.65)	Cysts were inoculated with excystation medium and incubated at 37 °C for 1–2 h. After centrifugation (200 g/10 min), the cysts were washed twice with distilled water and incubated with TYI-S-33 at 37 °C in an inclined position. The medium was changed daily for 3 days and subcultures were performed when the pH dropped below 6.4.	[50]
	1992	Faecal (human and animal)	Method of Sauch	Mixture of HCl–saline (concentrated HCl and 0.85% NaCl, pH 1.4) and HBSS (containing 32 mM glutathione with 57 mM L-cysteine, and 0.1 M NaHCO_3_)	A cyst suspension was mixed with excystation medium and incubated for 45 min at 37 °C. After centrifugation (650 g/2 min), the pellet was resuspended in Tyrode salt solution. An aliquot was removed and examined microscopically, while the remainder of the suspension was inoculated into prewarmed modified TYI-S-33 medium.	[37]
	1992	Mongolian gerbils	Method of Belosevic et al.	Not applicable	None-infected gerbils were gavage-fed 20 mg of metronidazole daily for 3 days and given dexamethasone (20 mg in 5 mL of PBS). A suspension of cysts in distilled water was inoculated into a gerbil, and 5–7 days later, the gerbil was sacrificed. The small intestine was removed, irrigated with prewarmed TYI-S-33, and the mucosa was scraped. The collected material was pooled and examined microscopically.	[37]
	1991	Cysts produced in vitro	Modified methods of Boucher and Gillin, Buchel et al., Schupp et al.	12 mM cysteine HCl/ascorbic acid in HBSS containing 1.500 U pepsin/mL	Cysts were incubated with excystation medium at 37 °C for 30 min. The medium was then neutralised by the adding 1 N NaHCO_3_ and centrifuged (1000 g/5 min). The resulting pellet was resuspended in TYI-S-33 and incubated at 37 °C. The cysts exposed to the cysteine/ascorbic acid solution (with or without pepsin) were subsequently incubated for 30 min at 37 °C with 1 mg/mL α-chrymotrypsin I-S in Tyrode’s solution (pH 8.0). The excystation medium was then replaced with TYI-S-33, and tubes were incubated at 37 °C and periodically examined for the presence of trophozoites	[39]
	1987	Faecal (human)	Method of Rice and Schaefer	Aqueous HCl (pH 2.0) mixed with 25 mL of HBSS (supplemented with 29 mM L-cysteine hydrochloride and 67 mM glutathione); and 2.5 mL of 0.1 M NaHCO_3_ 0.5% (wt/vol) trypsin (1:100) dissolved in Tyrode solution (pH 8.0)	A cyst suspension was incubated with a low-pH induction mixture for 30 min at 37 °C. After centrifugation (1000 g/2 min), the cysts were resuspended in excystation medium and incubated in an inverted position at 37 °C.	[29]

Caption: CaCl_2_—Calcium Chloride; HBSS—Hank’s Balanced Salt Solution; HCl—Hydrogen Chloride; KCl—Potassium Chloride; NaCl—Sodium Chloride; NaHCO_3_—Sodium Bicarbonate; NaOH—Sodium Hydroxide; PBS—Phosphate-Buffered Saline; USA—United States of America.

**Table 6 microorganisms-13-01719-t006:** Methods for the axenization of *G. lamblia* trophozoites.

Study Type	Year	Sample	Method	Culture Medium	Description	Reference
Experimental	2014	Reference isolate (ATCC 30888)	Modified method of Keister [56]	TYI-S-33	Trophozoites were maintained under axenic culture in culture medium.	[53]
	2007	Reference isolate (ATCC 30957) Laboratory isolate (STP)		Modified TYI-S-33 (pH 7.0), supplemented with 10% heat inactivated foetal calf serum and 0.05% bovine bile	The tubes were filled to 90–95% of total capacity with culture medium and incubated at 37 °C, with subculturing 3 times a week.	[19]
	2003	Faecal (human)		Modified TYI-S-33	Trophozoites were cultured and axenized in culture medium.	[26]
	1995	Faecal (human)		TYI-S-33	Axenic cultures were obtained by multiple washes with culture medium containing decreasing amounts of amphotericin B (10, 5, 2 to 1 µg/mL) by centrifugation (200 g/10 min). The cultures were established once no contamination (fungal—Sabouraud agar; bacterial—MacConkey agar, blood agar, thioglycolate broth) was observed.	[22]
	1992	Faecal (human and animal)		Filter-sterilised TYI-S-33 containing 500 IU/mL penicillin G, 50 µg/mL streptomycin, and 10 µg/mL amphotericin B	Excysted material was inoculated with culture medium at 37 °C.	[37]
	1991	Cysts produced in vitro		Filter-sterilised TYI-S-33 (pH 7.1) containing 10% bovine serum; 0.5 mg/mL of bovine bile; 100 U/mL of penicillin; and 0.1 mg/mL of streptomycin	Trophozoites were maintained axenically by subculturing twice a week and were grown to confluence (usually 66–72 h) at 37 °C.	[30]
	1991	Faecal (human)		TYI-S-33	After 5 days of excystation, subculturing was conducted with fresh culture medium. During growth, trophozoites formed a layer, which was dislodged by immersion in ice-water for 15 min and centrifuged (600 g/10 min/4 °C). The trophozoites were resuspended in 5 mL of fresh medium.	[15]
	1985	Duodenal fluid (human)	Method of Gordts et al. [19]	TPS-1 with trypticase peptone, liver digest, glucose, L-cysteine, and ascorbic acid, supplemented with 10% foetal calf serum, 10% NCTC-135 with L-glutamine, and a mix of antibiotics (penicillin (100 IU/mL), streptomycin (100 mg/l), vancomycin (20 mg/l), and clindamycin (20 mg/l))	Duodenal fluid was incubated with culture medium at 37 °C and screened for living trophozoites after 2 h. Cultures were inspected daily for 10 days.	[35]

**Table 7 microorganisms-13-01719-t007:** Methods for the encystation of *G. lamblia*.

Study Type	Year	Sample	Method	Encystation Medium	Description	Reference
Experimental	2017	Reference isolate (ATCC 50803)	Method of Barash et al.	Modified TYI-S-33 medium (pH 7.8) supplemented with 0.5 g/L bovine bile, 5% adult bovine serum, and 5% foetal bovine serum	Medium from 24 h cultures (~30% confluence) was decanted and replaced with the encystation medium. After 24 h, cysts were observed settled at the bottom of the tube.	[17]
			In vivo encystation	Not Applicable	Eight-week-old female C57/B6/J mice were orally gavaged with 1 × 10^7^ trophozoites in 100 µL phosphate-buffered saline.	
	1996	Reference isolate (ATCC 30957) Water (recreational and tap)	Modified method of Boucher and Gillin	Filter-sterilised TYI-S-33 (pH 7.8) supplemented with 5 mg/mL bovine bile, 0.6 mg/mL lactic acid hemi-calcium salt, and no antibiotics	Cultures were incubated at 37.5 °C for 66 h in encystation medium. The tubes were inverted eight times, with nonadherent cells decanted, pelleted, washed twice with double-distilled water, and incubated 30–45 min at room temperature to lyse trophozoites and immature cysts. Cysts were collected by low-speed centrifugation (135 g/5 min, 4–10 °C) and resuspended in sterile distilled water.	[23]
	1991	Reference isolates (ATCC 30888 and ATCC 30957) Laboratory strain (LT and WB clones) Cat strain	Method of Kane et al. [39]	Filter-sterilised TYI-S-33 medium (pH 7.8) with 10% bovine serum, 10 mg/mL bovine bile, 100 U/mL penicillin, and 0.1 mg/mL streptomycin	Confluent trophozoites were incubated at 37 °C for 96 h. After 24 h, cells were pelleted (500 g/10 min), resuspended in TYI-S-33 (pH 7.1), and further incubated. Cysts were collected by centrifugation (500 g/10 min) and washed twice with 20 mM sodium phosphate buffer (pH 7.1).	[39]
	1991	Reference isolates (ATCC 30888 and ATCC 30957) Laboratory strain (LT and WB clones) Cat strain	Production of cysts in roller bottles	Filter-sterilised TYI-S-33 medium (pH 7.8) with 10% bovine serum, 10 mg/mL bovine bile, 100 U/mL penicillin, and 0.1 mg/mL streptomycin	Trophozoites were grown to confluence in TYI-S-33 (pH 7.1) within roller bottles at 37 °C at 1 rotation/h. The medium was discarded; encystation medium was added and incubated at 37 °C for 24 h. The bottles were immersed in ice for 30 min and the contents were centrifuged (150 g/30 min). Encysting trophozoites were resuspended in TYI-S-33 and returned to the roller bottles for an additional 24 h at 37 °C. Cysts were harvested by immersion in ice for 30 min and centrifugation (150 g/30 min) then washed and incubated in distilled water.	[39]

## Data Availability

No new data were created or analysed in this study. Data sharing is not applicable to this article.

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
