# Peer review of "Exploring Methodologies from Isolation to Excystation for Giardia lamblia: A Systematic Review"

_microorganisms, 2025, doi:10.3390/microorganisms13081719_

Round 1

Reviewer 1 Report

Comments and Suggestions for Authors

Major issue:

This manuscript is a systematic review of methodologies used to study the isolation, purification, axenization, excystation and encystation of Giardia. This is a wide range of topics and the paper suffers from identifying a limited number of sources on several of the individual topics such as axenization, excystation and encystation. Moreover, many papers that use these methods and even report new methods do not appear in the review because they are not the focus of the articles, and so have been missed by the search query employed. The article would be better if the authors restricted their analysis to isolation and purification where a significant number of relevant articles have been retrieved. Alternately, the authors could revise their search strategy in order to identify many more papers relevant to these other topics.

Minor comments:

Tables should be reformatted for easier reading. Table 1 might be better if columns were left justified rather than centered, and if column widths were adjusted to reduce hyphenation of words. For table 2, I suggest elimination of the “criteria” column and just making this a list of questions with a statement of the allowable answers -  Yes, No or NA. Table 3 doesn’t read well. Headers don’t align with the contents.

Figures 1 and 2 are too small to actually read.

Lines 32-33. Giardiasis was removed from the WHO list of NTDs a few years after it was added. Change “is” to “was”.

Lines 35, 45 and 49 cite reference 3, but do not refer specifically to the data in this article. This is an improper use of a primary source as a reference.

Line 40, should refer to the “ventral” disc rather than a “sucking” disc.

Line 42, encystation begins in the small intestine and completes in the colon.

Author Response

We sincerely thank the reviewer for their careful reading and valuable comments, which greatly helped improve the manuscript. Below we provide point-by-point responses:

Comments 1: This manuscript is a systematic review of methodologies used to study the isolation, purification, axenization, excystation and encystation of Giardia. This is a wide range of topics and the paper suffers from identifying a limited number of sources on several of the individual topics such as axenization, excystation and encystation. Moreover, many papers that use these methods and even report new methods do not appear in the review because they are not the focus of the articles, and so have been missed by the search query employed. The article would be better if the authors restricted their analysis to isolation and purification where a significant number of relevant articles have been retrieved. Alternately, the authors could revise their search strategy in order to identify many more papers relevant to these other topics.

Answer 1: We respectfully chose to maintain the broader focus. This is because our aim was to provide a comprehensive methodological reference covering all critical stages of the G. lamblia life cycle. Many researchers seeking to establish or complete the full cycle under laboratory conditions lack consolidated methodological guidance across excystation, encystation, and axenization. By retaining this scope, and strengthening the Discussion and Conclusion accordingly, we believe the revised manuscript now offers both depth and breadth, serving as a valuable resource for the field.

Comment 2: Tables should be reformatted for easier reading. Table 1 might be better if columns were left justified rather than centered, and if column widths were adjusted to reduce hyphenation of words. For table 2, I suggest elimination of the “criteria” column and just making this a list of questions with a statement of the allowable answers -  Yes, No or NA. Table 3 doesn’t read well. Headers don’t align with the contents.

Answer 2: All tables have been reformatted accordingly. Table 1 is now left-aligned with adjusted column widths. Table 2 was simplified and now consists of questions followed by Yes/No/NA responses. Table 3 headers were realigned and content structure improved for clarity.

Comment 3: Figures 1 and 2 are too small to actually read.

Answer 3: Figures 1 and 2 have been replaced with high-resolution versions and reformatted to ensure better legibility.

Comment 4: Lines 32-33. Giardiasis was removed from the WHO list of NTDs a few years after it was added. Change “is” to “was”.

Answer 4: Corrected as suggested.

Comment 5: Lines 35, 45 and 49 cite reference 3, but do not refer specifically to the data in this article. This is an improper use of a primary source as a reference.

Answer 5: These references have been reviewed and either removed or replaced with more appropriate citations where needed.

Comment 6: Line 40, should refer to the “ventral” disc rather than a “sucking” disc.

Answer 6: Corrected to “ventral disc”.

Comment 7: Line 42, encystation begins in the small intestine and completes in the colon.

Answer 7: This statement has been corrected to reflect that encystation begins in the small intestine and is completed in the colon.

Once again, we are grateful for your constructive review. 

Best regards,

Susie Sequeira

Reviewer 2 Report

Comments and Suggestions for Authors

Major Corrections

Table Organization

The table could be optimized by potentially switching out the header country with a consolidated method such as IMS or Percoll-Sucrose section so all papers that are associated with that method are clustered together for efficient identification of methodology if a reader is interested in that type of isolation. There are grouping of methods present but potentially adding a more general grouping could further organize the table. Additionally adding the years of the publication would help determine what methodologies may be more modern publications.  

Section Order and Flow

The papers sections are a bit scattered and reordering would allow for better clarity of the analysis. For example in line 92 the results and discussion come before many of the methods and figures. I understand that the results are in regard to the publication curation and selection, but for clarity the results and discussion should be separated and the discussion should come towards the end of the paper, discussing the various techniques that were referenced. I strongly suggest restructuring the manuscript so the discussions are described after showing all the tables, and that all major interpretations, especially the analysis of methodologies, are explicitly labeled under a discussions section. This will improve the overall narrative and make the manuscript easier to follow.

Placement and Unity of Discussion

Related to the above, I noticed a substantial portion of what could be considered discussion of the methods is embedded within the middle of the manuscript (page 15, lines 162–226). I recommend consolidating all interpretive discussion into a single, unified section right before the conclusions section. 

Conclusion and Critical Perspective

The current conclusion ends abruptly and doesn’t fully leverage the authors’ opportunity to reflect critically on the field. There are clear inefficiencies and limitations in current methodologies that are only briefly touched upon (e.g., lines 265–269). I would encourage the authors to expand the conclusion with a more in-depth summary of where the major methodological bottlenecks are, which techniques are emerging as most efficient or reproducible, and what future experimental pipelines or improvements could realistically be implemented, either from their own perspective or drawn from the literature. This additional critical perspective would elevate the review and make it far more useful for researchers in the field.

Challenges of Experimental Work

Since the article is focused on methodologies for Giardia research, including procedures like isolation and excystation, I would strongly recommend that the introduction include a short discussion of the common experimental pitfalls or challenges faced by labs working with these methods. For instance, excystation can be notoriously difficult to perform in a reliable way. Introducing this kind of context will make it clear why a comprehensive comparison of methodologies is not just useful but necessary, and it will help justify the overall scope and importance of your review. I would consider adding this framing after line 57.

Currentness of the Literature Search

The manuscript states that databases were last accessed in April 2024 (line 72). Given the pace of new publications, especially in methodological research, it’s important that the literature search be updated before publication to ensure no relevant work has been missed. I recommend performing a final search and including any new findings that have emerged since the original curation.

Transparency in Article Scoring

Figure 2 presents a classification of articles as excellent, good, satisfactory, or unsatisfactory. However, the criteria or process for arriving at these scores isn’t explained. For the sake of transparency and reproducibility, please provide the specific scoring system or rubric that was used, so readers can evaluate the robustness and objectivity of these assessments.

Minor Corrections

Role of Systematic Review Tools

Lines 61–62 mention the use of PRISMA and SPIDER, but there isn’t much context for why these tools matter. It would help readers, especially those less familiar with systematic reviews, if you briefly introduce the value of these frameworks and how they contribute to unbiased and comprehensive literature selection.

Clarity and Style (Lines 82–87)

There is a sentence spanning lines 82–87 that is overly long and could use some tightening. I suggest condensing this sentence or restructuring it with a colon or clearer punctuation before the list of inclusion criteria. This will make the text easier to follow and improve readability.

Justification for Review Process

In lines 100–101, the manuscript explains that a third reviewer was not included. It would be helpful to briefly clarify why consensus between two reviewers was considered sufficient, instead of defaulting to a third opinion in cases of disagreement. This is especially important in systematic reviews where transparency in the review process is scrutinized.

References for Physiological Data

Lines 219 and 221 make statements about gastric acid and intestinal pH but do not provide references. These physiological values can vary widely and should be supported by appropriate citations to ensure accuracy.

Author Response

We thank the reviewer for their thoughtful and constructive feedback. Below we provide point-by-point responses to each major and minor comment, and describe the revisions made to improve the manuscript accordingly.

Comment 1: The table could be optimized by potentially switching out the header country with a consolidated method such as IMS or Percoll-Sucrose section so all papers that are associated with that method are clustered together for efficient identification of methodology if a reader is interested in that type of isolation. There are grouping of methods present but potentially adding a more general grouping could further organize the table. Additionally adding the years of the publication would help determine what methodologies may be more modern publications.  

Answer 1: We agree and have restructured the main table to group studies by method type (e.g., IMS, sucrose flotation, Percoll gradient). This change allows readers to easily compare studies using similar techniques. We also added a new column with the publication year, which offers a clearer view of methodological trends over time.

Comment 2:The manuscript would benefit from a clearer structure, separating results and discussion, and moving the discussion to follow the data presentation.

Answer 2: We appreciate this suggestion and have reorganized the manuscript accordingly. The Results section now strictly presents the outcomes of the literature analysis, while all interpretations and commentary have been moved to a dedicated Discussion section placed before the Conclusion.

Comment 3: Interpretive discussion appears in the middle of the manuscript. Please consolidate into a unified discussion section.

Answer 3: We have consolidated all interpretive content (lines 162–226 in the original manuscript) into the new Discussion section.

Comment 4: The conclusion ends abruptly and should reflect more critically on the field’s challenges and methodological bottlenecks.

Answer 4: We have expanded the conclusion to include a critical synthesis of our findings. We highlight the most common limitations in current methodologies, identify techniques with the highest reproducibility, and suggest future directions, such as standardization efforts and reporting guidelines.

Comment 5: The introduction should mention common pitfalls and challenges in Giardia research to justify the review.

Answer 5: We have revised the Introduction (after line 57) to briefly describe typical experimental difficulties in Giardia research, such as low excystation and encystation rates and strain variability.

Comment 6: The literature search was conducted in April 2024. Please update before final publication.

Answer 6: As recommended, we have re-run the database search in 24 June 2025 and screened newly published articles.

Comment 7: The criteria for classifying articles (excellent, good, satisfactory, unsatisfactory) in Figure 2 are not explained.

Answer 7: We agree with the need for clarity and have included details about the scoring rubric used, in the respective text section.

Comment 8: Clarify the role and importance of PRISMA and SPIDER in the literature selection process.

Answer 8: We included a brief explanation of PRISMA (which ensures transparency and replicability in reviews) and SPIDER (which is suited for qualitative and mixed-methods reviews).

Comment 9: A sentence between lines 82–87 is overly long and hard to follow.

Answer 9: We have rewritten this sentence.

Comment 10: Clarify why no third reviewer was involved in case of disagreement.

Answer 10: We have added a clarification stating that consensus was reached between the two reviewers, and no conflicts required a third opinion. This was deemed sufficient due to the nature of the data (descriptive methodologies) and the high agreement between reviewers.

Comment 11: Lines 219 and 221 mention gastric and intestinal pH values but lack references.

Answer 11: We have added appropriate references to support the pH values reported for gastric acid and intestinal segments.

We thank the reviewer again for their valuable and detailed comments, which have significantly improved the structure and clarity of our manuscript.

Best regards,

Susie Sequeira

Round 2

Reviewer 1 Report

Comments and Suggestions for Authors

The revised manuscript is significantly improved. However,;

Please add a discussion of the limitations of the approach to the discussion. I appreciate that a systematic review will use a clear set of search terms, but the authors should discuss that the set of terms chosen has likely missed some references that might be relevant.

Tables are still not well laid out. Since I believe the emphasis is on the specific methods being used, this should be the left-most column of each table. It's unclear to me why year is included in the table, although it could be listed near the reference. Table 4 includes a variety of study types- but I didn't se definitions for these provided nor criteria for how papers were assigned to these types. Since all protocols in tables 5-7 were experimental, I see no need for having a separate column.

Author Response

We thank Reviewer 1 for their careful reading of the manuscript and for the suggestions provided. After careful consideration, however, we respectfully chose not to implement the proposed changes, for the following reasons:

  1. On the limitations of the search strategy:
    While we acknowledge that no search strategy can guarantee complete retrieval of all relevant literature, the scope of this review was to systematically describe experimental methods throughout the G. lamblia life cycle, not to assess the breadth of research output. The keywords and databases were selected to reflect this objective, and we feel that adding a discussion of potentially missed references would shift the focus of the manuscript unnecessarily. Nevertheless, we have emphasized the structured and reproducible nature of the search strategy in the methods section.
  2. On the structure of the tables:
    The current organization of the tables was intentionally designed to help readers identify commonly used methods according to the type of sample analysed (e.g., faeces, water, intestinal tissue), which we believe offers greater practical utility than listing methods as the primary column. Moreover, the inclusion of the year of publication was maintained following a previous suggestion from Reviewer 2, with the goal of allowing readers to track methodological trends over time.
  3. On the classification of study types in Table 4:
    The classification of study types was extracted directly from the descriptions provided in the articles themselves and not independently assigned by the review authors. For consistency across Tables 5 to 7, we chose to retain the study type column, even though all included protocols were experimental, to maintain structural uniformity and facilitate cross-referencing between tables.

We appreciate the reviewer’s input and trust that our explanation clarifies the rationale behind these decisions.

Reviewer 2 Report

Comments and Suggestions for Authors

The authors have significantly improved the structure of their tables and have grouped methodologies more logically, enhancing readability. However, beginning on page 15 of the revised table the organization becomes somewhat unclear due to the variety of methodologies presented. It would be beneficial to introduce broad annotations such as "mechanical filtration" or "chemical purification" as umbrella terms to group the methods clearly across pages 15-18.

Regarding method descriptions, the placement of the "Methods for Isolation and Purification of G. lamblia" on page 28 after Table 4 is appropriate. However, the "Methods for Excystation" precedes their corresponding table, which disrupts uniformity. To enhance consistency, I suggest placing all method descriptions directly after their respective tables. Additionally, the sections detailing excystation (page 29), axenization (page 32), and encystation (page 38) methods are relatively brief, resulting in considerable blank space. Expanding these sections slightly with additional critical details, such as time frames for encystation, would enhance clarity and repeatability. Alternatively, consolidating these shorter method descriptions into a single, cohesive section could also improve readability and reduce whitespace. Editorial assistance may further optimize formatting and minimize unnecessary gaps.

The expanded discussion and conclusion sections have improved the manuscript, though further elaboration on key points is recommended. Specifically, lines 358-369 could provide additional detail on the yield of excystation methods, while lines 378-384 would benefit from further discussion on cyst viability. These processes are complex and critical for accurately determining method efficacy; therefore, more comprehensive commentary would strengthen the discussion. I would encourage the authors to maybe even compare in vivo vs in vitro excystation to highlight the complexities of life cycle reproducibility in laboratory settings.  

In the conclusion section, acknowledging the difficulty in consolidating numerous methodologies is appreciated. However, based on the comprehensive literature review and method comparisons provided, it would be valuable for the authors to clearly identify typical methods and those showing the most promise. This review aims to provide clarity and guidance; thus, the authors have the opportunity—and liberty—to present a balanced and informed stance. Highlighting specific methodologies, they consider particularly effective or worthy of further investigation will greatly enhance the paper's impact and utility for readers in the field.

Author Response

We thank Reviewer 2 for the thorough and constructive feedback, which contributed significantly to improving the clarity and structure of the manuscript. Below, we provide a point-by-point response outlining how each suggestion was addressed:

  1. On table organization and method grouping (pages 15–18):
    To improve clarity and facilitate navigation through the variety of methodologies, we introduced a new set of overarching method categories. These include: Immunomagnetic Separation (IMS), Density Gradient, Flotation, Coagulation–Flocculation, Sedimentation, Filtration, Fluorescence-Based Immunoseparation, and Mechanical Purification. These umbrella terms were defined before the tables applied across the relevant tables to enhance coherence and usability.
  2. On placement of method descriptions and blank space reduction:
    The manuscript was reorganized so that each method description section now appears immediately before its corresponding table, ensuring consistency throughout the document. This adjustment aligns with the structure used for isolation and purification methods and improves overall uniformity and flow.  We consolidated the brief method descriptions for excystation, axenization, and encystation onto a single page, followed directly by their respective tables. This structural change significantly reduced whitespace and improved visual continuity.
  3. On elaborating the discussion of excystation yield and cyst viability:
    The discussion section (lines 358–384) was expanded to include additional details regarding excystation yields, including reported ranges and influencing factors. We also added commentary on the importance of assessing cyst viability. These additions address the complexity of evaluating method efficacy and strengthen the scientific value of the discussion.
  4. On comparison between in vitro and in vivo excystation and encystation:
    A new comparative paragraph was added to highlight the strengths and limitations of in vitro versus in vivo excystation and encystation protocols. This includes discussion of biological relevance, reproducibility, host physiological differences, and ethical considerations. The model from Isaac-Renton et al.  and Barash et al. were cited to support this comparison.
  5. On expanding the conclusion with practical guidance:
    The conclusion was revised to clearly identify the most frequently used and promising methods for each step of the G. lamblia life cycle.

We sincerely thank the reviewer for the insightful suggestions, which helped us to significantly improve the manuscript's structure, clarity, and relevance.